# Modified mRNA-Mediated *CCN5* Gene Transfer Ameliorates Cardiac Dysfunction and Fibrosis without Adverse Structural Remodeling

**DOI:** 10.3390/ijms25116262

**Published:** 2024-06-06

**Authors:** Min Ho Song, Jimeen Yoo, Do-A Kwon, Elena Chepurko, Sunghye Cho, Anthony Fargnoli, Roger J. Hajjar, Woo Jin Park, Lior Zangi, Dongtak Jeong

**Affiliations:** 1College of Life Sciences, Gwangju Institute of Science and Technology, Gwangju 61005, Republic of Korea; songbese@naver.com (M.H.S.);; 2Cardiovascular Research Institute, Icahn School of Medicine at Mount Sinai, New York, NY 10019, USA; jimeenyoo@gmail.com (J.Y.); chepurkoelena@gmail.com (E.C.); fargnoli2@gmail.com (A.F.); 3Department of Medicinal & Life Science, College of Science and Convergence Technology, Hanyang University-ERICA, Ansan-si 15588, Republic of Korea; wldnjs9859@hanyang.ac.kr (D.-A.K.); josunghye@hanyang.ac.kr (S.C.); 4Mass General Brigham Gene and Cell Therapy Institute, Boston, MA 02139, USA; rhajjar@me.com

**Keywords:** modified mRNA (modRNA), CCN5, myocardial infarction (MI), cardiac fibrosis, heart failure (HF), left ventricular rupture

## Abstract

Modified mRNAs (modRNAs) are an emerging delivery method for gene therapy. The success of modRNA-based COVID-19 vaccines has demonstrated that modRNA is a safe and effective therapeutic tool. Moreover, modRNA has the potential to treat various human diseases, including cardiac dysfunction. Acute myocardial infarction (MI) is a major cardiac disorder that currently lacks curative treatment options, and MI is commonly accompanied by fibrosis and impaired cardiac function. Our group previously demonstrated that the matricellular protein CCN5 inhibits cardiac fibrosis (CF) and mitigates cardiac dysfunction. However, it remains unclear whether early intervention of CF under stress conditions is beneficial or more detrimental due to potential adverse effects such as left ventricular (LV) rupture. We hypothesized that CCN5 would alleviate the adverse effects of myocardial infarction (MI) through its anti-fibrotic properties under stress conditions. To induce the rapid expression of CCN5, ModRNA-*CCN5* was synthesized and administrated directly into the myocardium in a mouse MI model. To evaluate CCN5 activity, we established two independent experimental schemes: (1) preventive intervention and (2) therapeutic intervention. Functional analyses, including echocardiography and magnetic resonance imaging (MRI), along with molecular assays, demonstrated that modRNA-mediated *CCN5* gene transfer significantly attenuated cardiac fibrosis and improved cardiac function in both preventive and therapeutic models, without causing left ventricular rupture or any adverse cardiac remodeling. In conclusion, early intervention in CF by ModRNA-*CCN5* gene transfer is an efficient and safe therapeutic modality for treating MI-induced heart failure.

## 1. Introduction

Acute myocardial infarction (MI) is a major cause of morbidity and mortality in developed countries, with patient populations surpassing three million worldwide and causing more than one million deaths in the United States annually [1]. MI-induced HF is accompanied by deteriorated cardiac contractility and excessive cardiac remodeling [2,3]. During MI, ischemic cell death induces a multiphase reparative response in which the damaged tissue is substituted with fibrotic matrix deposition generated by fibroblasts and myofibroblasts (MyoFBs) [4]. Especially, MyoFBs transdifferentiated from FBs are the major cell types to produce excessive ECM and establish fibrosis in MI [4,5]. Since cardiac fibrosis (CF) is related to increased ventricular stiffness, cardiac dysfunction, arrhythmia, and impaired coronary blood flow [6,7,8,9,10], CF is a potential therapeutic target for cardiac dysfunction [11,12,13].

The ECM is an intricate network filling up the intercellular space. In the normal heart, cardiomyocytes compose the majority of the cardiac area with the ECM taking up less space than cells [14]. The ECM mechanistically supports the dynamic pumping of the heart and provides a pathway for signals to deliver into cells [15]. In MI, cardiomyocytes, which have extremely limited regenerative capacity in adult hearts, undergo structural changes. The infarcted ventricle repairs by clearing the wound of dead cells and matrix debris and the ECM restores the structural integrity of the ventricle [16,17,18]. Thus, suppressed ECM synthesis causes wall thinning at the infarct area and LV dilation occurs, which is possibly correlated with LV rupture. Some studies demonstrated that inhibition of fibrosis generates ventricular rupture and even unexpected mortality [19,20,21,22,23,24]. For example, ECM proteins, Periostin, and SPARC (Secreted Protein Acidic and Rich in Cysteine) deletion mouse models previously showed a lower survival rate due to ventricular rupture in the MI model [22,24]. Therefore, an anti-fibrotic treatment without LV rupture became one of the critical issues to overcome for therapeutic application under cardiac stress conditions.

The matricellular protein CCN5, also known as WNT1-inducible signaling pathway protein 2 (WISP-2), is a member of the cell communication network (CCN) family [25,26,27,28]. The CCN family is involved in various cellular processes, such as proliferation, differentiation, and migration. Our group previously reported that CCN5 prevents CF by inhibiting the pro-fibrotic protein connective tissue growth factor (CTGF), also known as CCN2 [29]. In addition, CCN5 reverses pre-established CF in two distinct ways: (1) inhibition of MyoFB transdifferentiation from FB; and (2) induction of MyoFB-specific apoptosis [30]. However, it is not clear whether early intervention in CF by *CCN5* gene transfer may induce LV rupture and affect animal mortality in the setting of HF.

Therefore, in the present study, we performed two different sets of experiments: (1) preventive intervention, in which modified mRNA-*CCN5* (ModRNA-*CCN5*) was injected at the time of MI, and outcomes were analyzed a week later; and (2) therapeutic intervention, in which ModRNA-*CCN5* was injected 2 weeks after MI, and the therapeutic effects were analyzed 2 weeks after injection. For accurate evaluation, echocardiography and MRI were performed to measure changes in cardiac remodeling and function. In addition, molecular signatures were investigated by immunohistochemistry, Western blotting, and quantitative real-time polymerase chain reaction (qRT-PCR). Collectively, ModRNA-*CCN5* gene transfer successfully inhibited cardiac remodeling by reducing CF, which, in turn, maintained cardiac function in both preventive and therapeutic models without LV rupture and adverse cardiac remodeling. In conclusion, early intervention in CF using CCN5 overexpression is a promising therapeutic modality for MI-induced HF.

## 2. Results

The primary aim of this study was to evaluate whether the early intervention of CF by ModRNA-*CCN5* gene transfer affects LV rupture and animal mortality. The secondary aim was to characterize the long-term therapeutic efficiency of ModRNA-*CCN5* treatment in the MI model. For the pilot echocardiography and MRI studies on the preventive intervention model, a total of 18 mice were used. Out of these mice, six animals died (two out of six from the ModRNA-Con group and four out of eight from the ModRNA-*CCN5* group). In most cases, the animal died immediately after intramyocardial injection with modRNAs, presumably due to the physical stress of the injection itself. No further mortality was observed after this point. Therefore, we conclude that acute CCN5 upregulation does not induce LV rupture or abnormal cardiac structural remodeling. For the echocardiography and molecular analyses, as well as the MRI study on the therapeutic intervention model, further animal experiments were conducted. The detailed animal numbers are described in each figure legend and table.

### 2.1. Preventive Intervention of ModRNA-CCN5 Attenuates MI-Induced Cardiac Dysfunction

To examine the effect of ModRNA-*CCN5* gene transfer on cardiac function, we induced MI in 8-week-old mice. Immediately after MI surgery, either ModRNA-Con or ModRNA-*CCN5* was directly injected into the myocardium of the mice in each group. One week later, echocardiography and MRI were performed to evaluate changes in cardiac function, and subsequently, the hearts were harvested for further characterization, as shown in Figure 1A. The ModRNA-Con-injected mice (n = 4) showed increased LVIDd and LVIDs compared to the sham-operated animals (n = 3). In contrast, the ModRNA-*CCN5*-injected mice showed a minimal increase in LVIDd and LVIDs (Figure 1B,C, n = 4). In addition, fractional shortening (FS), a systolic functional parameter, was significantly higher in the ModRNA-*CCN5*-injected group than in the ModRNA-Con-injected group. (Figure 1C).

Furthermore, a cardiac MRI was performed to accurately measure function and structural remodeling. The MRI results showed distinct differences among the three groups (Table 1). In terms of left ventricle ejection fraction (LVEF), one of the major functional parameters was substantially decreased in the ModRNA-Con group compared to the sham-operated group; however, the ModRNA-*CCN5*-injected mice showed significantly improved values than those of the ModRNA-Con group (Figure 1D). Moreover, LV mass and infarct size were substantially increased in the ModRNA-Con group, which were normalized in the ModRNA-*CCN5*-injected mice (Figure 1D). Taken together, preventive intervention using ModRNA-*CCN5* preserved cardiac function and structure after MI.

### 2.2. Preventive CCN5 Treatment Ameliorates MI-Induced CF without LV Rupture

After measuring cardiac function using echocardiography and MRI, additional mouse experiments were performed for further functional and molecular analyses (n = 6 for sham, n = 7 for ModRNA-Con, and n = 9 for ModRNA-*CCN5*). To determine the anti-fibrotic effect of ModRNA-*CCN5* in MI mice, we performed immunohistochemistry and other molecular studies, including Western blotting and qRT-PCR. Moreover, the cryopreserved cardiac tissues were stained with Masson’s trichrome reagent to analyze fibrotic and infarct areas post-MI. Additionally, *wheat germ agglutinin* (WGA) antibody was used to measure cell size. The result shows that the fibrotic and infarct areas were prominently increased in the ModRNA-Con-treated mouse hearts compared to those in the sham-operated mice. In contrast, the ModRNA-*CCN5*-injected mice showed a significantly reduced area of collagen deposition and infarction (Figure 2A). The fibrotic area over the remote area was measured using ImageJ software (Version 1.51) and is summarized in Figure 2B. Immunohistochemistry using the WGA antibody clearly shows the plasma membrane of cells; thus, the cell size can be measured accurately, as previously reported [31,32]. As a result, the ModRNA-Con-treated group showed enlarged cardiomyocytes, which was prevented by ModRNA-*CCN5* gene transfer (Figure 2A,C).

To further investigate the anti-fibrotic effect of CCN5, we analyzed the changes in fibrotic markers at the molecular level using Western blotting and qRT-PCR. Frozen cardiac tissues were homogenized to obtain proteins and mRNAs. The protein expression levels of FAP, Fibulin-1, *α*-SMA, and TGF-β1 were quantified by Western blotting. All pro-fibrotic markers were significantly elevated in the ModRNA-Con group (n = 7), but this elevation was inhibited by ModRNA-*CCN5* gene transfer (Figure 2D, n = 9). cDNA was also synthesized from mRNA and subjected to qRT-PCR analysis. The mRNA expression levels of *α-Sma*, *Tgf-β1*, and *Fap* were consistent with the Western blot analysis results. Additionally, *Bnp*, a major hypertrophic marker, was also significantly increased in the ModRNA-Con group; however, this increase was attenuated in the ModRNA-*CCN5*-injected group (Figure 2E). In conclusion, CCN5 prevents MI-induced CF without adverse cardiac remodeling or LV rupture.

### 2.3. Therapeutic Intervention of ModRNA-CCN5 Mitigates MI-Induced Cardiac Dysfunction

We further examined the therapeutic effects of ModRNA-*CCN5* on CF in a mouse model of MI. MI was produced on 8-week-old mice. Two weeks later, echocardiography was performed to confirm cardiac dysfunction. Subsequently, all mice were randomly distributed into two groups for further modRNA treatment. ModRNA-Con (n = 7) or ModRNA-*CCN5* (n = 8) were directly injected into the left ventricle of each group of mice. Two weeks post-injections, additional echocardiography and MRI were performed to investigate the change in cardiac function (Appendix A), followed by harvesting of heart specimens for histological and molecular analysis (Figure 3A).

Functional analysis indicated that LVIDs and LVIDd were not restored to the degree of the sham-operated group (n = 7), but they were substantially reduced in the ModRNA-*CCN5*-injected group than in the ModRNA-Con-injected group (Figure 3B). Moreover, FS showed a small but significant increase in the ModRNA-*CCN5*-injected group, which suggests that cardiac function was markedly improved (Figure 3C).

The following day, MRI was performed to assess the accurate cardiac structure and function. In line with the echocardiography data, ModRNA-*CCN5* treatment showed improved LVEF (Table 2), and notably, infarct areas were markedly reduced in the same group (Figure 3D, Appendix A).

Based on these data, we concluded that CCN5 treatment using modRNA has high therapeutic potential even under long-term cardiac stress conditions.

### 2.4. Therapeutic Intervention with ModRNA-CCN5 Reduces MI-Induced Cardiac Dysfunction and Fibrosis

After functional analysis, we also conducted immunohistochemistry and other molecular analysis to evaluate CF and HF marker gene expression. Firstly, cryopreserved cardiac tissues were stained with Masson’s trichrome reagent to analyze fibrotic and infarct areas after MI. Furthermore, *wheat germ agglutinin* (WGA) antibody was used to measure the individual cell size. Notably, similar results were obtained for the preventive intervention regimen. Fibrotic and infarct areas were distinctly increased in the ModRNA-Con-treated mouse hearts; however, they were substantially reduced in the ModRNA-*CCN5*-treated group (Figure 4A,B). In addition, the WGA staining results showed that the average cell size (CSA) in the ModRNA-*CCN5*-injected group was reduced compared to that in the ModRNA-Con-injected group, indicating inhibition of the hypertrophy phenotype (Figure 4C).

Moreover, we decided to further characterize the additional markers for fibrosis, calcium signaling, and cardiac stress to convince our functional data, especially for the therapeutic intervention regimen. Firstly, in order to validate the presented cardiac function, calcium handling proteins, such as SERCA2a, NCX1, and p-PLB/PLB, and cardiac stress markers such as ANP were also measured by Western blotting. Secondly, we evaluated multiple pro-fibrotic markers such as TGF-β1, p-Smad2, Fibronectin, Periostin, α-SMA, and Vimentin. The protein expression levels of SERCA2a and p-PLB, but not NCX1, were significantly restored by *CCN5* gene transfer (Figure 4D). Also, as shown in Figure 4E, the levels of all pro-fibrotic markers were markedly elevated in the ModRNA-Con group, which were inhibited by ModRNA-*CCN5* gene transfer. More importantly, ANP expression was substantially downregulated; thus, it reflected the anti-fibrotic and anti-hypertrophic effects of ModRNA-*CCN5* gene transfer (Figure 4E). Notably, CCN5 expression itself is not much different among all groups because modRNA expression is usually sustained for up to 10 days, which is in line with the previous literature [33]. Finally, the qRT-PCR analysis also showed similar results, in that all inflammatory and fibrotic markers, such as *Tnf-α*, *Mcp-1*, *Il6*, *Tgf-β1*, *α-Sma*, *Mmp2*, and *Fap*, were significantly reduced (Appendix A). In contrast, the anti-inflammatory marker, *Il10*, was substantially increased in the ModRNA-*CCN5*-delivered group; thus, these results were in line with the Western blot analysis data. Furthermore, the *Bnp* mRNA level was also considerably attenuated by *CCN5* delivery (Appendix A). In conclusion, the therapeutic intervention of ModRNA-*CCN5* also successfully reduced cardiac dysfunction and remodeling by inhibiting inflammation and CF.

## 3. Discussion

mRNA-mediated gene transfer has regained attention during the recent COVID-19 pandemic as a safe and efficient delivery method to treat human diseases. For the last decade, adeno-associated virus (AAV)-mediated gene transfer has been favored due to its high transfection efficiency and relatively minimal immune response. However, the risk of insertional mutagenesis and induction of unexpected immune responses still remains a critical concern for their safe application. Previously, mRNA-mediated gene transfer was proposed as a safe non-viral delivery method, since mRNA exerts its function in the cytoplasm; thus, the risk of insertional mutagenesis was excluded. Moreover, repeated applications are also possible, which is the biggest limitation of viral vector-mediated gene transfer. Lastly, viral vector-induced immune responses can also be avoided [34].

Unfortunately, in vitro transcribed mRNA shows strong immunogenicity to TLR3, TLR7, and TLR8, a major obstacle to its successful application in vivo [35,36]. Despite these benefits, therefore, the use of mRNA as a delivery method for gene therapy has been hampered for many years. However, recent reports and the development of nucleic acid modification techniques have opened a new era for utilizing mRNA as an efficient carrier for the gene of interest. Although further development of mRNA is required for the treatment of chronic inherited diseases, mRNA-mediated gene therapy is a promising modality for the treatment of various human diseases.

CF is a major complication of HF. Since CF is highly related to muscle stiffness and elastance in cardiac function, it has been a therapeutic target to treat HF [11,12,13]. However, there is controversy regarding anti-fibrotic treatment to cure HF, because fibrosis in the early stage of cardiac stress conditions is necessary to maintain the cardiac structure [37,38]. Moreover, LV rupture is a critical hurdle for anti-fibrotic therapy in the MI model based on the previous literature [19,20,21,22,23,24], which was not the focus of our previous research [30].

CCN5 has been proposed as a promising therapeutic target for resolving CF induced by HF [29,30]. Our group has shown three major signaling pathways to support the activity of CCN5: (1) inhibition of the TGF-β signaling pathway, (2) inhibition of transdifferentiation from fibroblasts (FBs) into MyoFBs, and (3) induction of specific apoptosis in MyoFBs. However, early intervention in CF was not elucidated in a previous study since we used AAV9-mediated gene transfer. Therefore, we utilized modRNA-mediated gene transfer to generate a rapid CCN5 expression in the present study. As shown in the results section, *CCN5* gene transfer using modRNA did not induce any adverse remodeling of the cardiac structure, including LV rupture, in both the preventive and therapeutic studies. Moreover, it clearly improves cardiac function.

Based on recent reports, cardiac rupture after MI has been identified due to excessive regional inflammation [39,40]. In the ischemic damaged area, myocardial necrosis induces activation of complement, generation of free radicals, and the cytokine cascade. Thus, neutrophils are recruited to the ischemic area and exert potent cytotoxic effects by releasing proteolytic enzymes that affect cardiomyocytes [38]. In this regard, our group has shown in a previous study that CCN5 exhibits anti-inflammatory activity [41,42]. For example, *CCN5* knockout mice highly expressed pro-inflammatory genes in mice with lipotoxic cardiomyopathy [41]. In addition, *CCN5* administration mitigated the upregulation of pro-inflammatory genes in an angiotensin II-infused mouse model [42]. Therefore, the anti-inflammatory activity of CCN5 may also contribute to the reduction in CF without LV rupture in the MI model. Although most mortality in this study was observed during the modRNA injection procedure for cardiac delivery, a superior cardiotropic delivery method should be developed for further in vivo approaches and potential clinical applications. In fact, many applications have recently been developed, such as oral delivery with nanoparticle-mediated modRNA or aerosolized modRNA delivery [43,44]. Therefore, the disadvantages of direct injection into the heart may soon be overcome. Conclusively, we demonstrated that anti-fibrotic treatment using modRNA-mediated *CCN5* gene transfer is a highly promising therapeutic approach for the treatment of MI-mediated HF.

## 4. Materials and Methods

### 4.1. Animal Care and Myocardial Infarction

All experimental procedures were approved by the Animal Care Committee of the Gwangju Institute of Science and Technology (approval number: GIST-2020-104) and the Institutional Animal Care and Use Committee of the Icahn School of Medicine at Mount Sinai (protocol number: IACUC-2017-0200). The C57BL/6 mice were obtained from The Jackson Laboratory (Bar Harbor, ME, USA) and Orient Bio (Seongnam-si, Gyeonggi-do, Republic of Korea). Male C57Bl/6 mice aged 8–10 weeks (weight, 25–30 g) were ventilated under anesthesia with isoflurane. The chest was opened before ligation of the left anterior descending coronary artery and closed after the operation was completed.

### 4.2. Production of Modified mRNA-CCN5 and Injection

Polyadenylated and capped (CleanCap™ technology) modRNAs were synthesized by Trilink Biotechnologies (San Diego, CA, USA) with full substitution of uridine with pseudo-uridine, following DNase treatment. The 5′ and 3′ UTRs were designed by Zangi lab (New York, NY, USA). Below is the ORF sequence of ModRNA-*CCN5*

ATGAGAGGCACACCGAAGACCCACCTCCTGGCCTTCTCCCTCCTCTGCCTCCTCTCAAAGGTGCGTACCCAGCTGTGCCCGACACCATGTACCTGCCCCTGGCCACCTCCCCGATGCCCGCTGGGAGTACCCCTGGTGCTGGATGGCTGTGGCTGCTGCCGGGTATGTGCACGGCGGCTGGGGGAGCCCTGCGACCAACTCCACGTCTGCGACGCCAGCCAGGGCCTGGTCTGCCAGCCCGGGGCAGGACCCGGTGGCCGGGGGGCCCTGTGCCTCTTGGCAGAGGACGACAGCAGCTGTGAGGTGAACGGCCGCCTGTATCGGGAAGGGGAGACCTTCCAGCCCCACTGCAGCATCCGCTGCCGCTGCGAGGACGGCGGCTTCACCTGCGTGCCGCTGTGCAGCGAGGATGTGCGGCTGCCCAGCTGGGACTGCCCCCACCCCAGGAGGGTCGAGGTCCTGGGCAAGTGCTGCCCTGAGTGGGTGTGCGGCCAAGGAGGGGGACTGGGGACCCAGCCCCTTCCAGCCCAAGGACCCCAGTTTTCTGGCCTTGTCTCTTCCCTGCCCCCTGGTGTCCCCTGCCCAGAATGGAGCACGGCCTGGGGACCCTGCTCGACCACCTGTGGGCTGGGCATGGCCACCCGGGTGTCCAACCAGAACCGCTTCTGCCGACTGGAGACCCAGCGCCGCCTGTGCCTGTCCAGGCCCTGCCCACCCTCCAGGGGTCGCAGTCCACAAAACAGTGCCTTCTAG

ModRNA and Lipofectamine RNAiMAX transfection reagent (Invitrogen, Waltham, MA, USA, #13778150) were separately diluted in Opti-MEM (Invitrogen, Waltham, MA, USA, #31985070), mixed, and then incubated for 15 min at room temperature. An amount of 0.5 μL of RNAiMAX reagent was used for 1 μg of ModRNA. The mixture was directly injected into the endocardium nearby the infarcted area. An amount of 100 μg of naked ModRNA-Con or -*CCN5* was used for each heart.

### 4.3. Echocardiography

Echocardiography was performed under sedation with the intraperitoneal injection of ketamine (100 mg/kg). Sedation was optimized by (1) providing the lowest dose of ketamine needed to restrain the animals and to prevent motion artifacts and (2) maintaining the heart rate as close as possible to 550 beats/min. Two-dimensional images and M-mode tracings were recorded on the short axis at the level of the mid-papillary muscle to determine percent fractional shortening and ventricular dimensions (GE Vivid 7 Vision) as recommended by the American Society of Echocardiography.

### 4.4. Magnetic Resonance Imaging

Delayed-enhancement CINE images were obtained on a 7-T Bruker Pharmascan with cardiac and respiratory gating (SA Instruments, Morrisville, NC, USA). For imaging, the mice were anesthetized with 1–2% isoflurane in the air. To monitor optimal temperature during the ECG, respiratory and temperature probes were placed on the mouse. Imaging was performed 10–20 min after IV injection of 0.3 mmol/kg gadolinium-diethylene triamine pentaacetic acid. A stack of 8–10 short-axis slices spanning from the heart apex to its base was acquired with an ECG-triggered and respiratory-gated FLASH sequence with the following parameters: echo time (TE) 2.7 ms with 200 μm × 200 μm resolution; 1 mm slice thickness; 16 frames per R-R interval; and 4 excitations with a flip angle of 60°. After imaging, the obtained data were analyzed to calculate % ejection fraction, cardiac output, stroke volume, and % MI size.

### 4.5. Masson’s Trichrome Staining and *Wheat Germ Agglutinin* Staining

The murine cardiac tissues were cryopreserved with Tissue-Tek OCT Compound (Sakura, St. Torrance, CA, USA, #4583) and sliced into 6 µm sections. The tissues were fixed with 4% paraformaldehyde and stained with Masson’s trichrome or fluorescein isothiocyanate-labeled *wheat germ agglutinin* (WGA). Fibrotic areas were visualized using a ZEISS Axiophot microscope (Dublin, OH, USA), and WGA fluorescent signals were analyzed using a ZEISS Axio Imager.D2. The percentage of fibrotic area was calculated as the ratio of the total area of fibrosis to the total area of the section. Cell surface area (CSA) was calculated by measuring the size of the WGA-stained plasma membranes. ImageJ was used to measure the fibrotic area and CSA.

### 4.6. Western Blot Analysis

Cardiac tissue lysates were solubilized with RIPA buffer (50 mM Tris, 150 mM NaCl, 0.1% sodium dodecyl sulfate, 1% Triton X-100, pH 8.0), and protease inhibitor cocktail set III (Merck Millipore, Darmstadt, Germany, #535140) was added. Protein amounts in the solubilized tissue lysates were quantified using the Pierce BCA Protein Assay Kit (Thermo Scientific, Waltham, MA, USA, #23227). Quantified tissue proteins were separated by sodium dodecyl sulfate–polyacrylamide gel electrophoresis and transferred to polyvinylidene difluoride membranes (Merck Millipore, Darmstadt, Germany, #IPVH00010). The membranes were blocked with 5% non-fat skim milk and incubated with anti-SERCA2a (Badrilla, Leeds, UK, #A010-23L), anti-α-SMA (Sigma-Aldrich, St. Louis, MO, USA, #A5228), anti-CCN5 (c-term, MyBioSource, San Diego, CA, USA, #MBS3216315), anti-Vimentin (Cell Signaling Technology, Danvers, MA, USA, #5741), anti-Fibronectin (Proteintech, Rosemont, IL, USA, #15613-1-AP), anti-Periostin (Invitrogen, Waltham, MA, USA, #PA5-34641), anti-TGF-β (Cell Signaling Technology, Danvers, MA, USA, #3711), anti-p-Smad2 (pSer465, 467, Invitrogen, Waltham, MA, USA, #MA5-15122), anti-Smad2/3 (Cell Signaling Technology, Danvers, MA, USA, #5678), anti-NCX1 (Invitrogen, Waltham, MA, USA, #MA3-926), p-PLB (pSer16, Badrilla, Leeds, UK, A010-12AP), PLB (Badrilla, Leeds, UK, A010-14), anti-TGF-β1 (Abcam, Cambridge, UK, #ab215715), anti-ANP (Abcam, Cambridge, UK, #ab262703), anti-BNP (Abcam, Cambridge, UK, #ab239510), and anti-GAPDH (Santa Cruz, CA, USA, #sc-47724) antibodies for 12–16 h at 4 °C. The membranes were rinsed with Tris-buffered saline containing 0.1% Tween 20 (TBS-T), incubated with horseradish peroxidase (HRP)-conjugated secondary antibodies (Thermo Scientific, Waltham, MA, USA, #31430), and rinsed again. The band signal was developed using Immobilon Western Chemiluminescent HRP substrate (Millipore, Darmstadt, Germany, #WBKLS0500).

### 4.7. Quantitative Real-Time Polymerase Chain Reaction

Total RNA was isolated from cardiac tissue using an RNA purification kit (Qiagen, Hilden, Germany, #74004). cDNA was synthesized using ImProm-II reverse transcriptase (#A3802; Promega, Madison, WI, USA) with oligo-dT priming. PCR was performed using a Thermal Cycle Dice Real-Time System TP800 (Takara, San Jose, CA, USA) with TB Green Premix (Takara, San Jose, CA, USA, #RR820A) as the fluorescent dye. The primer sequences used in this study are listed in Appendix A.

### 4.8. Statistics

Student’s *t*-test and one-way analysis of variance were used to determine the significance of the data. A single asterisk (*) corresponds to *p* < 0.05, a double asterisk (**) corresponds to *p* < 0.01, and a triple asterisk (***) corresponds to *p* < 0.001. Data are presented as mean ± standard deviation.

## Figures and Tables

**Figure 1 ijms-25-06262-f001:**
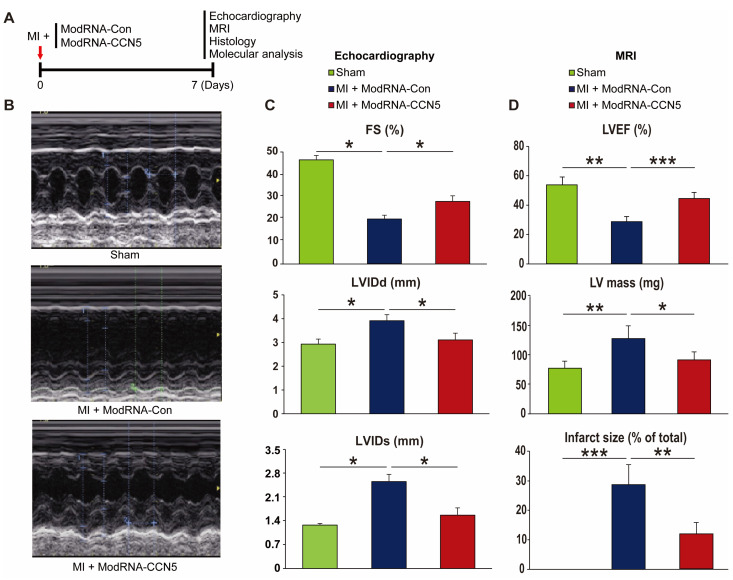
Preventive intervention of ModRNA-*CCN5* attenuates MI-induced cardiac dysfunction. (**A**) MI was induced in the mouse heart and subsequently, ModRNA-Con or -*CCN5* was directly injected into the LV endocardium of the mouse. A week later, echocardiography, histology, and molecular analysis were performed to determine the effects of modified mRNA-*CCN5*. (**B**) Representative M-mode echocardiographic images are shown. (**C**) Fractional shortening (FS), LVIDd, and LVIDs were compared, sham (n = 6) vs. MI + ModRNA-Con (n = 7) vs. MI + ModRNA-*CCN5* (n = 9). (**D**) The results of the MRI analysis are shown. Left ventricular ejection fraction (LVEF), LV mass, and infarct size were compared, sham (n = 3) vs. MI + ModRNA-Con (n = 4) vs. MI + ModRNA-*CCN5* (n = 4). * *p* < 0.05, ** *p* < 0.01, *** *p* < 0.001.

**Figure 2 ijms-25-06262-f002:**
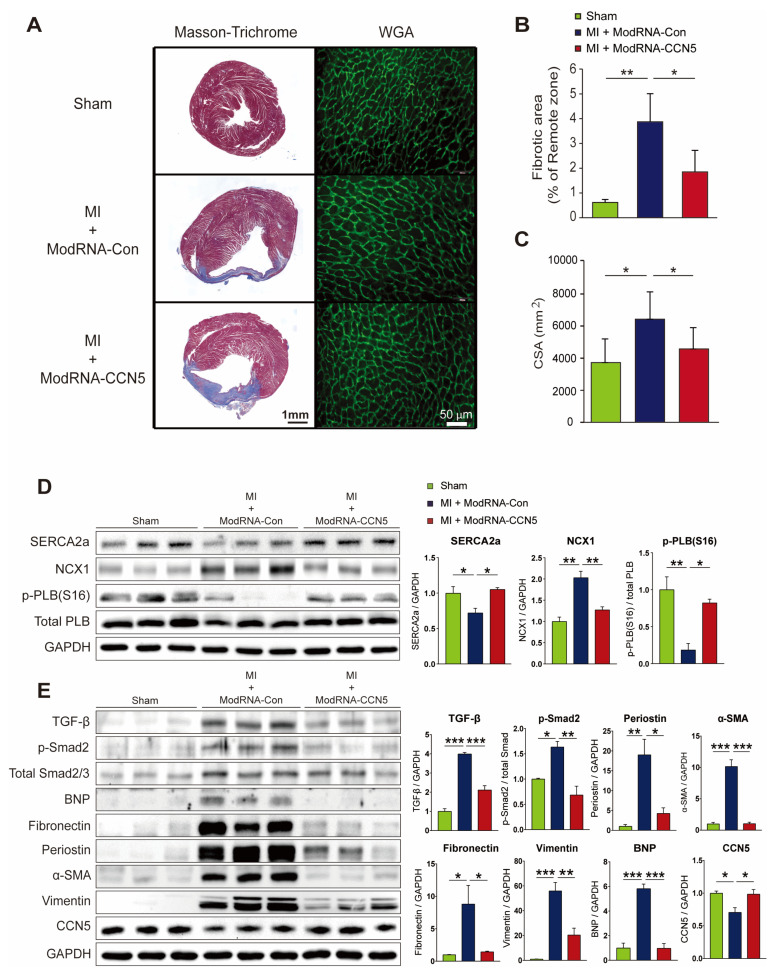
Preventive CCN5 treatment ameliorates MI-induced cardiac fibrosis without LV rupture. (**A**) Heart tissues from the preventive intervention were used for all experiments. The red-stained area indicated normal tissue and the blue-stained area indicated fibrotic tissue in Masson’s trichrome staining. WGA staining was performed to measure the cellular size of the heart. The green-stained area indicated a cellular membrane. Scale bar: 50 μm. (**B**) Fibrotic area was analyzed using a computer-assisted method. ImageJ plug-in software was used to measure the ratio of fibrotic area. (**C**) Cell surface area (CSA) was analyzed using a computer-assisted method. ImageJ plug-in software was used to measure CSA. (**D**) Cardiac tissue lysates (n = 6 for sham, n = 7 for ModRNA-Con, and n = 9 for ModRNA-*CCN5*) were immunoblotted with antibodies against FAP, Fibulin-1, *α*-SMA, CCN5, GAPDH, and TGF-β1. (**E**) Synthesized cDNAs were used to analyze the mRNA expression level of *α-Sma*, *Tgf-β1*, *Fap*, and *Bnp* by qRT-PCR. * *p* < 0.05, ** *p* < 0.01, *** *p* < 0.001.

**Figure 3 ijms-25-06262-f003:**
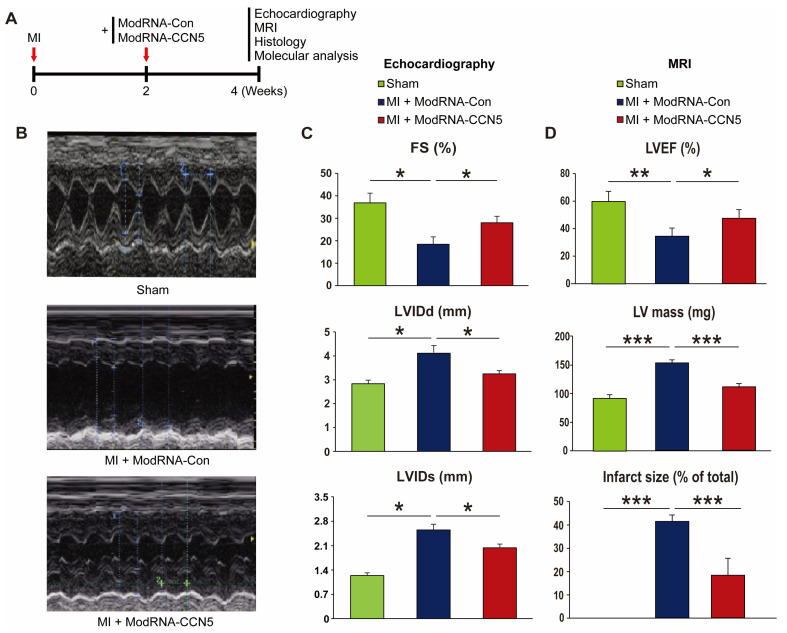
Therapeutic intervention of ModRNA-*CCN5* mitigates MI-induced cardiac dysfunction. (**A**) MI was induced in the heart. Then, 2 weeks later, ModRNA-Con or -*CCN5* was directly injected into the endocardium of the mouse LV (therapeutic intervention). Subsequently, 4 weeks later, echocardiography, MRI, histology, and molecular analysis were performed to determine the effects of modified mRNA-*CCN5*. (**B**) Representative M-mode echocardiographic images are shown. (**C**) Fractional shortening (FS), LVIDd, and LVIDs were compared, sham (n = 7) vs. MI + ModRNA-Con (n = 7) vs. MI + ModRNA-*CCN5* (n = 8). (**D**) MRI was performed in each group of mice (n = 3 for sham, n = 4 for ModRNA-Con, and n = 4 for ModRNA-*CCN5*) and some critical values are presented. Left ventricular ejection fraction (LVEF), LV mass, and infarct size were compared, sham vs. MI + ModRNA-Con vs. MI + ModRNA-*CCN5*. * *p* < 0.05, ** *p* < 0.01, *** *p* < 0.001.

**Figure 4 ijms-25-06262-f004:**
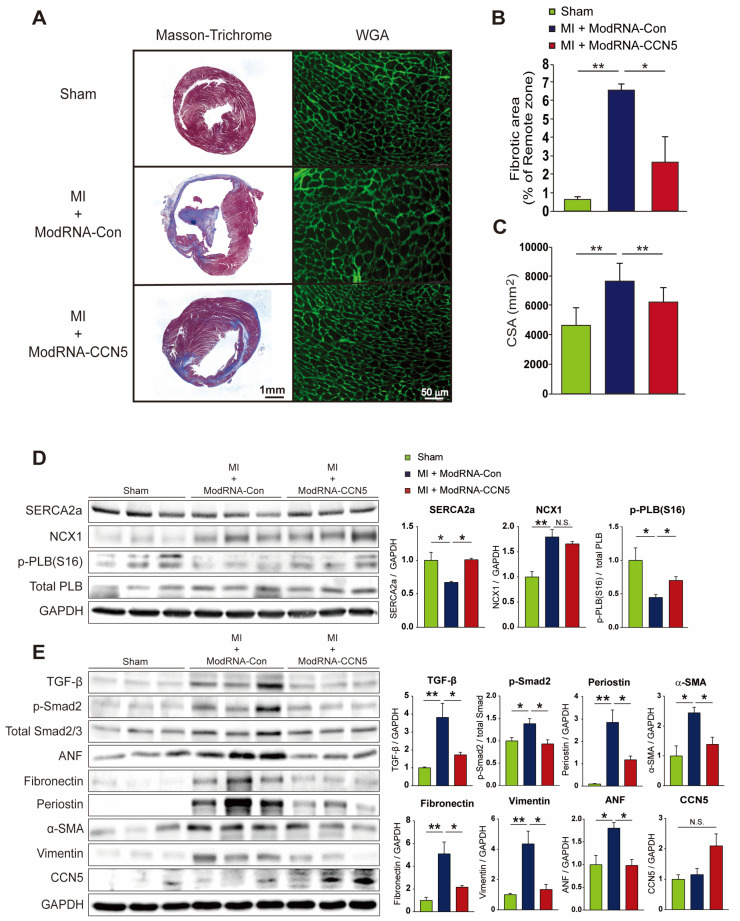
Therapeutic intervention with ModRNA-*CCN5* reduces MI-induced cardiac dysfunction and fibrosis. (**A**) The heart tissues from the therapeutic intervention were used for all experiments. The red-stained area indicated normal tissue and the blue-stained area indicated fibrotic tissue in Masson’s trichrome staining. WGA staining was performed to measure the cellular size of the heart. The green-stained area indicated a cellular membrane. Scale bar: 50 μm. (**B**) Fibrotic area was analyzed using a computer-assisted method. ImageJ plug-in software was used to measure the ratio of fibrotic area. (**C**) Cell surface area (CSA) was analyzed using a computer-assisted method. ImageJ plug-in software was used to measure CSA. (**D**) Cardiac tissue lysates were immunoblotted with antibodies against SERCA2a, NCX1, p-PLB (Ser 16), total-PLB, and GAPDH. (**E**) Additional lysates were immunoblotted with antibodies against TGF-β, p-Smad2, Smad2/3, ANF, Fibronectin, Periostin, α-SMA, Vimentin, CCN5, and GAPDH. For each experimental group, animals were used as follows: n = 7 for sham, n = 7 for ModRNA-Con, and n = 8 for ModRNA-*CCN5*, * *p* < 0.05, ** *p* < 0.01, n.s.: not significant.

**Table 1 ijms-25-06262-t001:** Cardiac MRI parameters for the preventive studies.

Preventive Intervention	Sham (n = 3)	MI + ModRNA-Con (n = 4)	MI + ModRNA-CCN5 (n = 4)
**LVEDD (mm)**	4.35 ± 0.37	5.93 ± 0.51	4.78 ± 0.53
**LVESD (mm)**	2.91 ± 0.46	4.97 ± 0.52	3.52 ± 0.25
**FS (%)**	33.1 ± 6.68	16.19 ± 4.47 *	26.35 ± 4.26 *
**LVEDV (μL)**	66.20 ± 7.82	117.25 ± 26.67 **	81.45 ± 18.28 **
**LVESV (μL** **)**	24.42 ± 1.03	89.42 ± 32.56 **	42.36 ± 17.13 **
**SV (μL** **)**	52.13 ± 8.05	40.00 ± 5.29	34.00 ± 5.29
**LVEF (%)**	55.25 ± 5.27	29.76 ± 3.54 ^**^	45.77 ± 4.27 **
**LV Mass (mg)**	82.31 ± 12.32	134.65 ± 22.47 *	96.85 ± 14.25 *
**LV Length (mm)**	6.62 ± 0.43	8.36 ± 0.42	7.17 ± 0.52
**LV Width (mm)**	3.07 ± 0.46	5.65 ± 0.35	3.73 ± 0.52
**Infarct Size (%)**	-	29.33 ± 6.78 **	12.24 ± 3.89 **

LVEDD: Left ventricular end-diastolic diameter, LVESD: Left ventricular end-systolic diameter, FS: Fractional shortening, LVEDV: Left ventricular end-diastolic volume, LVESV: Left ventricular end-systolic volume, SV: Stroke volume, LVEF: Left ventricular ejection fraction. * *p* < 0.05, ** *p* < 0.01.

**Table 2 ijms-25-06262-t002:** Cardiac MRI parameters for the therapeutic studies.

Therapeutic Intervention	Sham (n = 3)	MI + ModRNA-Con (n = 4)	MI + ModRNA-CCN5 (n = 4)
**LVEDD (mm)**	5.25 ± 0.37	6.27 ± 0.82	5.72 ± 0.48
**LVESD (mm)**	3.31 ± 0.62	5.06 ± 0.27	4.13 ± 0.54
**FS (%)**	37.24 ± 2.84	19.30 ± 5.83 **	27.78 ± 5.37 **
**LVEDV (μL)**	62.74 ± 3.66	127.45 ± 17.93 **	86.88 ± 3.64 **
**LVESV (μL** **)**	19.36 ± 4.34	91.43 ± 16.43 **	40.45 ± 6.71 **
**SV (μL** **)**	43.38 ± 6.26	36.02 ± 4.24	46.43 ± 3.83
**LVEF (%)**	60.25 ± 7.32	34.87 ± 5.87 **	47.86 ± 6.22 **
**LV Mass (mg)**	90.00 ± 6.37	148.62 ± 4.28	108.25 ± 5.85
**LV Length (mm)**	7.62 ± 0.56	8.54 ± 0.28	8.10 ± 0.72
**LV Width (mm)**	3.64 ± 0.72	6.35 ± 0.75	4.75 ± 0.34
**Infarct Size (%)**	-	41.27 ± 2.76 **	18.48 ± 7.25 **

LVEDD: Left ventricular end-diastolic diameter, LVESD: Left ventricular end-systolic diameter, FS: Fractional shortening, LVEDV: Left ventricular end-diastolic volume, LVESV: Left ventricular end-systolic volume, SV: Stroke volume, LVEF: Left ventricular ejection fraction. ** *p* < 0.01.

## Data Availability

All data that support the finding of this study are available from the corresponding author upon reasonable request.

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
