# Peer review of "Modified mRNA-Mediated CCN5 Gene Transfer Ameliorates Cardiac Dysfunction and Fibrosis without Adverse Structural Remodeling"

_ijms, 2024, doi:10.3390/ijms25116262_

Round 1

Reviewer 1 Report

Comments and Suggestions for Authors

IJMS- 3010136 comments

To extend their previous AAV-CCN5 therapy study, Song et al now used modified mNRA (modRNA)-mediated gene transfer in a mouse acute myocardial infarction (MI) model. In both the preventive and therapeutic intervention schemes, CCN5 mRNA treatment was able to inhibit cardiac fibrosis (CF) and improve cardiac dysfunction. Here are a few points to improve the manuscript.  

1.        Six experimental animals died in the study, with two from the modRNA-Con control and four from the modRNA CCN5 treatment groups. As a result, the animal number in each group was very small for statistical analysis. This needs to be discussed.

2.        Please follow the guideline for gene and protein nomenclature. All the genes need to be italic.  

3.        All the data presented in this manuscript were descriptive. It is still not clear how CCN5 functions in this MI model. To be published in IJMS, the authors need to include some mechanistic data, like the downstream signal pathways.

4.        CCNs are multimodular proteins, including the insulin-like growth factor binding protein (IGFBP) homology domain, a von Willebrand factor type C repeat domain (vWC), a thrombospondin type 1 repeat (TSP1) and a C-terminal cysteine-knot (CK) domain. The authors may consider characterize role of each CCN5 domain in this MI model.

5.        Animal numbers used for statistical analysis in each figure need to be mentioned in the figure legend.   

Author Response

First of all, I deeply appreciate all your insightful comments.

Please find the attached answers to your concerns below.

To extend their previous AAV-CCN5 therapy study, Song et al now used modified mNRA (modRNA)-mediated gene transfer in a mouse acute myocardial infarction (MI) model. In both the preventive and therapeutic intervention schemes, CCN5 mRNA treatment was able to inhibit cardiac fibrosis (CF) and improve cardiac dysfunction. Here are a few points to improve the manuscript.  

  1. Six experimental animals died in the study, with two from the modRNA-Con control and four from the modRNA CCN5 treatment groups. As a result, the animal number in each group was very small for statistical analysis. This needs to be discussed.

--> Thank you for the important comment. Also, we apologize the lack of all the details about the animal experiments.

Initially, we started the experiment by the MRI analysis for the preventive intervention models to get the primary insight and accurate functional measurement. Since MRI analysis was costly experiment, we used small number of animals as described in the result section (Total 18 mice, 4 for sham, 6 for modRNA-Con, and 8 for modRNA-CCN5).

However, after we obtained positive results from this pilot MRI study, we used extra animals for further functional and molecular analysis as follows.

- Preventive intervention: Sham (n=6), MI + modRNA-Cont (n=7), MI + modRNA-CCN5 (n=9)

- Therapeutic intervention; Sham (n=7), MI + modRNA-Cont (n=7), MI + modRNA-CCN5 (n=8)

- MRI study for the therapeutic intervention models; Sham (n=3), MI + modRNA-Cont (n=4), MI + modRNA-CCN5 (n=4)

All detailed information has been updated in the revised manuscript.

  1. Please follow the guideline for gene and protein nomenclature. All the genes need to be italic.

--> Thank you for your insightful comment. We have revised the manuscript as suggested accordingly.

  1. All the data presented in this manuscript were descriptive. It is still not clear how CCN5 functions in this MI model. To be published in IJMS, the authors need to include some mechanistic data, like the downstream signal pathways.

--> Previously, we have demonstrated that CCN5 ameliorates cardiac fibrosis through the inhibition of the TGF-beta signaling pathway and selective apoptosis on myofibroblast, a major cell type producing ECM molecules (JACC, 2016). In addition, we have also shown that CCN5 enhances cardiac function through the restoration of SERCA2a expression and Phospholamban (PLB) phosphorylation in another paper (FCVM, 2022).

Based on these previous results, as shown in Fig 2 and 4 in this manuscript, we also evaluated the expression of several calcium-regulating proteins including SERCA2a, NCX, and PLB. Those protein expressions were substantially restored after modCCN5 injection, contributing to better cardiac performance. However, we don’t rule out the involvement of other molecular mechanisms under CCN5 overexpression. In fact, recently we found a clue that CCN5 overexpression also reduced the miRNA-25 expression through the apoptotic regulation of myofibroblasts. Since miR-25 was reported to be a critical regulator of SERCA2s (Nature, 2014), it is one of the possible mechanisms to restore the SERCA2a expression.

We will perform further experiments to fully understand the whole picture of CCN5 activity including miRNA-25 regulation.

Once again, the major interest of this paper is whether the inhibition of fibrosis by CCN5 in the early stage of MI is beneficial or not. Therefore, we sharply focused on this aspect.

  1. CCNs are multimodular proteins, including the insulin-like growth factor binding protein (IGFBP) homology domain, a von Willebrand factor type C repeat domain (vWC), a thrombospondin type 1 repeat (TSP1) and a C-terminal cysteine-knot (CK) domain. The authors may consider characterize role of each CCN5 domain in this MI model.

--> This is a great suggestion.

In general, CCN5 lacks the CK domain, which is a major structural difference from other CCN families, such as CCN2 (CTGF). Since CCN2 is one of the most well-characterized pro-fibrotic molecules, we previously hypothesized that the CK domain is essential for the pro-fibrotic activity of CCN2. In fact, it was confirmed that a CK domain-deletion mutant of CCN2 did not show any profibrotic effects. Furthermore, fusion of the CK domain to CCN5 transformed CCN5 into CCN2-like pro-fibrotic molecule (JMCC,2010). Therefore, I am afraid that an additional domain study of CCN5 may be a redundant experiment.

  1. Animal numbers used for statistical analysis in each figure need to be mentioned in the figure legend.   

--> Thank you for your comment. All information has been updated in the figure legend.

References

  1. Matricellular Protein CCN5 Reverses Established Cardiac Fibrosis. J Am Coll Cardiol. 2016 Apr 5;67(13):1556-1568.
  2. Matricellular Protein CCN5 Gene Transfer Ameliorates Cardiac and Skeletal Dysfunction in mdx/utrn (±) Haploinsufficient Mice by Reducing Fibrosis and Upregulating Utrophin Expression. Front Cardiovasc Med. 2022 Apr 26;9:763544.
  3. Inhibition of miR-25 improves cardiac contractility in the failing heart. Nature. 2014 Apr 24;508(7497):531-5
  4. The opposing effects of CCN2 and CCN5 on the development of cardiac hypertrophy and fibrosis. J Mol Cell Cardiol. 2010 Aug;49(2):294-303.

Reviewer 2 Report

Comments and Suggestions for Authors

In this original research paper, Min Ho Song et al examined the contribution of modRNA-CCN5 gene transfer to the cardiac fibrosis after myocardial infarction in mice. 

The protocol design and execution of the study are meticulously performed. Three groups were compared (sham, intervention, control) with several imaging techniques and molecular biology techniques. The authors suggest that early intervention in cardiac fibrosis by modRNA-CCN5 gene transfer is an efficient and safe therapeutic modality for treating MI-induced heart failure without adverse structural remodelling. 

The main concern in this manuscript is the sample size; mice used in each group. How did the authors calculate the number of animals required for the study? Did they perform a power analysis?

Currently, in the results section it is not clear how many animals were included in the study. Also it is not clear how many animals were included in each group to perform statistics. It seems that 4/6 mice in the modRNA-Con, 4/8 in the modRNA-CCN5. It seems there is no information for this about the sham group. This is important to decide whether the statistic test (student's t test and one-way analysis) is appropriate. Also, in the bar graphs, what do the error bars represent? standard deviation or standard error?

The reviewer believes that this is an important study, well-executed. But further information on the sample size and statistical analysis is necessary to decide whether this manuscript is ready for publication. 

Author Response

I deeply appreciate all your comments.

Please find the attached answers to your concerns on the manuscript below.

In this original research paper, Min Ho Song et al examined the contribution of modRNA-CCN5 gene transfer to the cardiac fibrosis after myocardial infarction in mice. 

The protocol design and execution of the study are meticulously performed. Three groups were compared (sham, intervention, control) with several imaging techniques and molecular biology techniques. The authors suggest that early intervention in cardiac fibrosis by modRNA-CCN5 gene transfer is an efficient and safe therapeutic modality for treating MI-induced heart failure without adverse structural remodelling. 

The main concern in this manuscript is the sample size; mice used in each group. How did the authors calculate the number of animals required for the study? Did they perform a power analysis?

Currently, in the results section it is not clear how many animals were included in the study. Also it is not clear how many animals were included in each group to perform statistics. It seems that 4/6 mice in the modRNA-Con, 4/8 in the modRNA-CCN5. It seems there is no information for this about the sham group. This is important to decide whether the statistic test (student's t test and one-way analysis) is appropriate. Also, in the bar graphs, what do the error bars represent? standard deviation or standard error?

--> Thank you for the important comment. Also, we apologize for the lack of all the details about the animal experiments.

Initially, we started the experiment by the MRI analysis for the preventive intervention models to get the primary insight and accurate functional measurement. Since MRI analysis was costly experiment, we used small number of animals as described in the result section (Total 18 mice, 4 for sham, 6 for modRNA-Con, and 8 for modRNA-CCN5).

However, after we obtained positive results from this pilot MRI study, we used extra animals for further functional and molecular analysis as follows.

- Preventive intervention: Sham (n=6), MI + modRNA-Cont (n=7), MI + modRNA-CCN5 (n=9)

- Therapeutic intervention; Sham (n=7), MI + modRNA-Cont (n=7), MI + modRNA-CCN5 (n=8)

- MRI study for the therapeutic intervention models; Sham (n=3), MI + modRNA-Cont (n=4), MI + modRNA-CCN5 (n=4)

All detailed information has been updated in the revised manuscript.

Lastly, as mentioned in the 4.8. Statistics section, error bars present standard deviation.

The reviewer believes that this is an important study, well-executed. But further information on the sample size and statistical analysis is necessary to decide whether this manuscript is ready for publication. 

--> Once again, we are sorry for the lack of detailed information.

All information on the sample size has been updated in the revised manuscript.

Round 2

Reviewer 1 Report

Comments and Suggestions for Authors

I have no more questions.

Reviewer 2 Report

Comments and Suggestions for Authors

The reviewer would like to thank the authors for their kind reply and clarifications. Of course the set of experiments used in the study design are costly, which can be a limitation and I am sure that the authors know that this cannot be used as an excuse to compensate for appropriate sample size.

The edits to the figure legends and methods are important and are well taken into consideration.  

The reviewer would like to congratulate the authors for their elegant study and manuscript, which is worthy for publication.